# Occurrence of Fungi in the Potable Water of Hospitals: A Public Health Threat

**DOI:** 10.3390/pathogens9100783

**Published:** 2020-09-24

**Authors:** Giuseppina Caggiano, Giusy Diella, Francesco Triggiano, Nicola Bartolomeo, Francesca Apollonio, Carmen Campanale, Marco Lopuzzo, Maria Teresa Montagna

**Affiliations:** Department of Biomedical Science and Human Oncology-Hygiene Section, University of Bari Aldo Moro, Piazza G. Cesare 11, 70124 Bari, Italy; giusy.diella@uniba.it (G.D.); francesco.triggiano@uniba.it (F.T.); nicola.bartolomeo@uniba.it (N.B.); francesca.apo@libero.it (F.A.); campanale.carmen@libero.it (C.C.); marcolopuzzo@gmail.com (M.L.); mariateresa.montagna@uniba.it (M.T.M.)

**Keywords:** fungi, molds, potable water, hospitals, public health, *Aspergillus*

## Abstract

Since the last decade, attention towards the occurrence of fungi in potable water has increased. Commensal and saprophytic microorganisms widely distributed in nature are also responsible for causing public health problems. Fungi can contaminate hospital environments, surviving and proliferating in moist and unsterile conditions. According to Italian regulations, the absence of fungi is not a mandatory parameter to define potable water, as a threshold value for the fungal occurrence has not been defined. This study evaluated the occurrence of fungi in potable water distribution systems in hospitals. The frequency of samples positive for the presence of fungi was 56.9%; among them, filamentous fungi and yeasts were isolated from 94.2% and 9.2% of the samples, respectively. The intensive care unit (87.1%) had the highest frequency of positive samples. Multivariable model (*p* < 0.0001), the variables of the period of the year (*p* < 0.0001) and type of department (*p* = 0.0002) were found to be statistically significant, suggesting a high distribution of filamentous fungi in the potable water of hospitals. Further studies are necessary to validate these results and identify the threshold values of fungi levels for different types of water used for various purposes to ensure the water is safe for consumption and protect public health.

## 1. Introduction

Since the last decade, attention towards the occurrence of fungi in potable water has increased, and they are now considered possible contaminants responsible for causing health problems in humans. Fungi are commensal or environmental saprophytic microorganisms widely distributed in nature, which can cause opportunistic infections in immunocompromised individuals. To date, nosocomial mycoses are important causes of morbidity and mortality [1,2,3]; particularly, the bloodstream yeast infections caused by *Candida* spp. are increasing significantly worldwide, representing an important infective complication in hospitalized patients with medical and surgical disorders [4,5]. Similarly, mold infections pose serious health threats in particular to patients with hematological malignancies, especially during prolonged periods of neutropenia. These infections have also been recognized as emerging opportunistic infections in patients with chronic obstructive pulmonary disease, HIV, or those admitted to the intensive care unit [6,7,8]. The most common forms of molds causing serious infections are *Aspergillus*, *Mucorales*, *Fusarium*, and *Scedosporium* [2,9,10,11,12,13]. Fungi can contaminate hospital environments, and survive and proliferate in moist and unsterile conditions [14] such as construction, demolition, or accumulation of dust during cleaning activities. These conditions increase the dispersion of propagules of molds, increasing the exposure of patients to these opportunistic pathogens [14]. Therefore, to minimize microbial contamination through air, heating, ventilation, and air conditioning systems have been introduced in hospitals [15,16]. Fungi are natural inhabitants of water and can grow in dead spaces and pipelines (with low circulation) of water systems [17]. Stagnation of water allows the formation of a biofilm, which plays an important role in the accumulation, propagation, and dissemination of pathogens, including filamentous fungi [18]. Until a few years ago, the occurrence of fungi in water was rarely considered, because attention was directed towards occurrence of bacteria and viruses [19,20,21]. These microorganisms unlike fungi cause acute diseases when they contaminate potable water, and are now regarded as emerging causes of chronic water quality problems [22,23,24].

Fungal propagules become aerosolized when contaminated water passes through showerheads, taps, and toilets, and cause respiratory exposure to susceptible patients, especially in areas of major water use [14]. Anaisse et al. [25] found *Aspergillus* species in the hospital water system and verified that concentration of airborne *Aspergillus* propagules was significantly higher in bathrooms than in patient rooms and in hallways.

Studies have reported the occurrence of fungal communities in potable water [26] and downstream water distribution networks [27,28,29,30], despite conventional water treatment systems [31,32]. During the 1980s and 1990s, an increased number of cases with health problems caused by fungal-contaminated potable water were reported in Finland and Sweden [33,34]. This prompted an investigation on the occurrence of fungi in potable water in several countries [9,31,35]. Although the clinical risks associated with the exposure to yeasts and molds are recognized, especially for immunocompromised patients, according to the national regulation Italian Decree 31/01 [36] and Italian Decree 27/02 [37], occurrence of fungi has not been made a mandatory parameter to define potability of water, and to date, no threshold value for the occurrence of fungi has been defined.

Based on these reports and considering to the best of our knowledge that there are no Italian investigations on fungal contamination of hospital waters, we aimed to evaluate the occurrence of fungi in the potable water distribution networks of hospitals.

## 2. Results

The frequency of samples positive for fungi was 56.9% (207/364), with an average of 5.8 (range 1–15) cfu/100 mL (Figure 1). Of these, filamentous fungi were isolated from 94.2% (195/207) of samples and yeasts from 9.2% (19/207). Particularly, 90.8% (188/207) of the positive samples were contaminated by different species of filamentous fungi, 5.8% (12/207) were contaminated by yeasts, while 3.4% (7/207) were contaminated simultaneously by different species of both.

Among yeasts, *Rhodotorula* spp. were most frequently isolated (73.7%; 14/19), followed by *Cryptococcus* spp. (15.8%; 3/19) and *Candida* spp. (10.5%; 2/19) (Table 1).

Among filamentous fungi, *Fusarium* spp. were most frequently isolated (26.67%; 52/195), followed by *Aspergillus* spp. (21%; 41/195) and *Cladosporium* spp. (19%; 37/195) (Table 1). Besides, 6.3% (13/207) of the samples had multiple contaminations by three or more different strains; 16.9% (35/207) contained two strains and 76.8% (159/207) only one strain.

Intensive care unit was the type of ward (Table 2) with the highest frequency of positive samples (87.1%; 27/31), followed by the intermediate services (mainly laboratories) (82.6%; 19/23), surgery wards 57.1% (52/91), and medicine 49.3% (103/209). In other units (management office, service rooms, etc.), 60% (6/10) of the samples were positive. No statistical association was observed between the type of wards (chi-square = 29.94, *p* = 0.07) and type of room (chi-square = 19.3027, *p* = 0.2003).

Yeasts had a high prevalence in the intensive care unit samples (22.6%; 7/31), while molds were prevalent in samples from both intensive care unit (74.2%; 23/31) and intermediate services (82.6%; 19/23). The frequency of molds in the surgery and medicine wards were 53.8% (49/91) and 46.89% (98/209), respectively.

The risks associated with a positive sample were evaluated using the univariable logistic regression model. The model revealed a statistical significance associated with type of wards (LRT = 25.18, *p* < 0.0001). The chi-square test also revealed the statistical significance of the variable (chi-square = 19.27, df = 4, *p* = 0.0007).

Among the different types of rooms, (Table 3) 62.5% (65/104) positive samples were obtained from hospital rooms, followed by 57.6% (38/66) from bathrooms and 54.8% (40/73) from operating theaters. From these values, it is evident that the type of room was not statistically significant (LRT = 2.278, *p* = 0.5167). A statistical significance was found between the periods of sampling. When one year was split into three periods of 4 months each, 67% (77/115) of the samples obtained in the first, 48% (108/225) in the second, and 91.7% (22/24) in the third period were positive (Figure 1).

A multivariable logistic model was used to evaluate the variables found in higher frequencies in positive samples. The period of sample collection and type of department were used as explanatory variables. The type of room and its association with the type of department were removed from the model due to statistical insignificance.

The multivariable model (LRT = 54.3069, *p* < 0.0001), periods of the year (chi-square = 22.8794, *p* < 0.0001), and type of department (chi-square = 22.4785, *p* = 0.0002) showed statistically significant results. In particular, the contamination risk was found to be higher in the first quarter of the year compared to the second (OR = 2.94, 95% CI 1.7–5.09) and in the third quarter compared to the second (OR = 14.9, 95% CI 2.9–76.9) (Figure 2).

The intensive care unit showed a higher number of positive samples compared to the medicine (OR = 9.2, 95% CI 2.9–28.4), and surgery wards (OR = 4.4, 95% CI 1.4–13.9). The surgery ward showed a greater risk than the medicine department (OR = 2.09, 95% CI 1.2–3.6), and the intermediate services had a greater risk than the medicine department (OR = 3.2, 95% CI 1.08–10.2).

## 3. Discussion

Filamentous fungi and yeasts are ubiquitous organisms found in association with organic matter. They are mostly parasites and opportunistic pathogens capable of causing diseases in immunocompromised subjects. Despite this, little attention had been given to their presence and significance in aquatic environments [17].

In our study, more than half of the samples tested resulted positive for fungi and most often for a particular genus and species. According to the European Commission, water intended for human consumption should be “free from any microorganisms and parasites, and any substances which, in numbers or concentrations, constitute a potential danger to human health” [38]. However, as the concentration of fungi in water was not considered quantitatively relevant as that of bacteria and other microbial contaminants, it has led to the omission of detecting the occurrence of fungi in routine potable water quality monitoring tests, resulting in a lack of information on the possible impacts on human health [24,39].

The search for fungi in hospital water networks presents some limitations related to the lack of specific guidelines and contamination indicators that can be correlated with the possible presence of fungi. In fact, only in the Legislative Decree of February 2, 2002 n.27, [37] the search for fungi on 100 mL was included as accessory parameter but without any indication of limit values. Only in 2005, the Guidelines of the Italian Society of Nephrology, referring to the microbiological controls provided for dialysis waters, included the search for fungi distinguishing yeasts and molds and specifying reference values and frequency of investigation [40].

A key issue related to the occurrence of fungi in water is the presence of biofilms. The complex structures of biofilms are formed by the combined growth of fungi together with other microorganisms such as bacteria and protists, and are reservoirs of these microbes [41]. The fungal cells within a biofilm reveal a form of resistance, but the mechanisms that govern its growth are unclear [41].

Biofilm formation also affects the quality of water by reducing the effectiveness of residual chlorine, raising the costs of maintenance of potable water distribution networks [24]. This increased resistance to chlorine disinfection is likely due to the consumption of disinfectants by the biofilm, reduced disinfectant penetration into the biofilm, and composition of the microbial community in the biofilm itself [41].

One of the main problems related to the presence of fungi in drinking water is the poor effectiveness of disinfection treatments of the water network on some fungal genera. Dematiaceous fungi, like *Alternaria*, secreting melanin or melanin-like pigment in their thick cell walls, have hydrophobic spores, which make them resistant to water treatments [24,30,42,43].

Our results revealed the presence of *Fusarium*, *Aspergillus*, *Cladosporium*, *Acremonium*, *Penicillium*, *Paecilomyces*, *Alternaria*, *Candida*, and *Cryptococcus* in accordance with other studies [32,41].

These species are often dangerous for patients’ health, causing respiratory illness, cutaneous or systemic infections, both in adults and children [22,32,44]. Etienne et al. [45], investigating a cluster of cutaneous aspergilloses in a neonatal intensive care unit, reported that the humidity chambers of the neonates’ incubators could be the source of the infection by *A. fumigatus*.

Among yeasts, *Candida* is an important causative agent of cutaneous or invasive infections, associated with significant morbidity, prolonged hospital stays, high mortality, and increased healthcare costs [46]. Presence of *Candida* in aquatic sources may be due to contamination with human or animal excrement [47]. Although *Rhodotorula* species are considered typically environmental saprophytes, they can cause important infections associated with intravenous catheter, particularly in immunocompromised hosts. Cases of catheter-associated fungemia due to *R. mucilaginosa* in immunocompetent host are also reported [48].

In our study, the positivity for molds was significantly higher than that for yeasts (91.78% vs. 8.2%). Although literature data about mold behavior in the network are limited, and guidelines regarding acceptable mold values do not exist, it is important to note that health problems linked to fungi in water are also due to the production of secondary metabolites such as mycotoxins. Some molds such as, *Penicillium* spp., *Fusarium* spp., *Aspergillus* spp., can produce toxic secondary metabolites that prejudice water quality and become a threat to humans and animals [24,43,49]. Some of these products are carcinogenic and have the ability to impair the immune system [43,50,51]. Although the levels of mycotoxins in water are particularly low, their concentrations may increase to hazardous levels when water is stored in reservoirs for long periods and also be present in the treated drinking water long after the fungi has died [24,49].

Often some fungal strains can also produce other secondary metabolites that exude organic acids, which contribute to microbiological corrosion of water pipes [52]. Consequently, the increased metal concentration impedes proper water disinfection because of the difficulties in maintaining the sufficient concentrations of residual chlorine in the distribution system [43]. In the iron pipes, the products of corrosion react with residual chlorine and prevent it from penetrating the biofilm [43].

In this study, higher frequency of the positive samples was observed in the intensive care unit and laboratories, probably related to the infrequent use of tap water compared to that in the medicine and surgery departments, favoring the formation of biofilms. As there was no statistical difference, in occurrence of fungi, observed between the type of rooms, contamination in the whole water network and not only on the taps was confirmed.

The data suggest that the current treatments of potable water are ineffective on fungi, particularly the filamentous fungi. Hospital water networks are often old and proper disinfection of water without damaging pipes is difficult, and poses a great risk to the patients in the hospital as they are exposed to potable water, which meets the legislative criteria, but is actually potentially dangerous due to contamination by molds.

## 4. Materials and Methods

### 4.1. Study Design

Potable water was investigated for the occurrence of fungi between January 2018 and December 2019, using 364 tap water samples, as an accessory parameter, according to the Italian Legislative Decree 31/2001 [36]. The study was carried out in a large university hospital with 1400 beds that covers an area of approximately 230 thousand square meters in the Apulia region, in southern Italy. Taps were disinfected with a solution of 10% sodium hypochlorite, and flamed, then cold water was allowed to flow for 1–3 min before sampling.

### 4.2. Water Sampling and Mycology Investigation

Sampling and processing procedures were performed according to the national regulation Italian Legislative Decree (Decree 31/01). Cold water samples (1 L) were collected from taps in sterile containers with sodium thiosulphate pentahydrate (0.01%, *w/v*), to neutralize the chloride present in potable water, and transported at a controlled temperature of 4 °C in the dark. A 100 mL aliquot of the water sample was filtered through a cellulose ester double membrane (47 mm Ø and 0.45 μm-pore size, Millipore, Milan, Italy), placed on a Sabouraud dextrose agar medium containing chloramphenicol (0.5 g/L) (Liofilchem, Roseto degli Abruzzi, Italy), incubated at 30 ± 1 °C, and checked daily for growth for 8 days.

The enumeration of fungi was evaluated on five days of incubation and the results were expressed as colony forming units (cfu)/100 mL.

### 4.3. Fungal Identification

After sub-isolation onto specific media such as malt extract agar (Liofilchem, Roseto degli Abruzzi, Italy) and Czapek yeast extract agar (Liofilchem, Roseto degli Abruzzi, Italy) genus and species of the filamentous fungi isolates were identified based on their macroscopic and microscopic morphological features. In accordance with the methods described by de Hoog [53], the macroscopic examination was based on visual observation of morphological characteristics and color of aerial mycelium, while the microscopic analysis was performed by preparation of lactophenol cotton blue-stained slides. The slides were prepared with tape that adhered to aerial mycelium and placed on the lactophenol cotton blue-stained slides. Yeasts were identified by a semi-automated sugar assimilation system API ID32 (Biomérieux Italia S.p.A., Bagno a Ripoli, Italy).

### 4.4. Statistical Analysis

All data were analyzed using qualitative variables and presented as numbers and percentages. To evaluate the environmental characteristics that could be involved in the contamination, all operative units were classified as intensive care unit, medicine, surgery, intermediate services (most of all laboratories), and other services. The type of room, where the sample was collected was classified as baths, hospital rooms, operative rooms, and others (management office, service rooms etc.). The period of the year, when sampling was done was classified as January to April, May to August, and September to December.

The association between the sampling results and other variables was determined using the chi-square test. The effects of the main characteristics were evaluated using univariable and multivariable logistic regression models. A likelihood ratio test (LRT) was used to assess the significance of the model and type III analysis of effects in logistic regression was used to assess the significance of the variables. Odds ratio (OR) and 95% confidence interval (CI) for all comparisons were determined.

Data were analyzed using SAS 9.4 software and a *p*-value of < 0.05 was considered statistically significant.

## 5. Conclusions

This study highlights a high distribution of fungi, particularly filamentous fungi, in potable water of hospitals, posing a risk to the consumers, especially those who are immunocompromised. Therefore, it would be appropriate to extend the regulations to include a mandatory microbiological parameter for fungi as well. Further studies are necessary to validate this data and identify the threshold levels of fungi for different types of water with different scopes of consumption (hospital, domestic, industrial, etc.).

## Figures and Tables

**Figure 1 pathogens-09-00783-f001:**
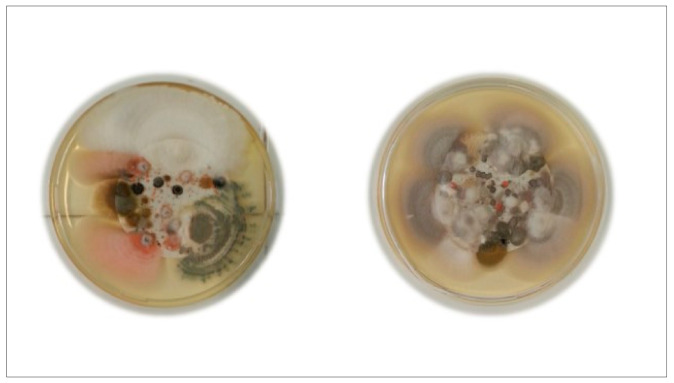
Sabouraud dextrose agar plates after eight days of incubation with cellulose ester membrane: different species of fungi from 100 mL of filtered water.

**Figure 2 pathogens-09-00783-f002:**
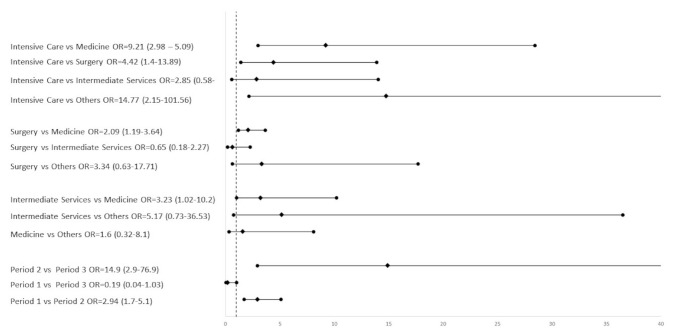
Odds ratio (95% confidence interval) to compare risk between type of wards and period of the year in which sample take place. The vertical dashed line is for odds ratio equal to 1, if the continuous line of the confidence interval crosses the dashed line, it means that there is no difference in risk between the groups compared. The diamond-shaped dot represents the estimate point, while round dots represent the confidence limits.

**Table 1 pathogens-09-00783-t001:** Frequency of species isolated by type of ward. Percentages are determined as the ratio between the number of isolated species and the total into bracket. Multiple isolation could happen.

Fungal Genera	Surgery (No = 52)		Intensive Care (No = 27)		Medicine (No = 103)		Intermediate Service (No = 19)		Others (No = 6)		Total (No = 207)	
	No	%	No	%	No	%	No	%	No	%	No	%
*Fusarium*	5	9.62	4	14.81	33	32.04	9	47.37	1	16.67	52	25.12
*Aspergillus*	12	23.08	7	25.93	19	18.45	3	15.79	0	-	41	19.81
*Cladosporium*	6	11.54	3	11.11	22	21.36	2	10.53	4	66.67	37	17.87
*Penicillium*	5	9.62	3	11.11	15	14.56	1	5.26	1	16.67	25	12.08
*Acremonium*	6	11.54	2	7.41	16	15.53	0	-	0	-	24	11.59
*Paecilomyces*	11	21.15	3	11.11	3	2.91	0	-	0	-	17	8.21
*Rhodotorula*	3	5.77	5	18.52	6	5.83	0	-	0	-	14	6.76
*Alternaria*	6	11.54	1	3.70	5	4.85	1	5.26	0	-	13	6.28
*Mucorales*	3	5.77	0	-	8	7.77	1	5.26	1	16.67	13	6.28
*Chrysosporium*	0	-	1	3.70	3	2.91	3	15.79	0	-	7	3.38
*Scedosporium*	1	1.92	2	7.41	1	0.97	0	-	0	-	4	1.93
*Scopulariopsis*	0	-	1	3.70	1	0.97	2	10.53	0	-	4	1.93
*Trichophyton*	1	1.92	0	-	2	1.94	1	5.26	0	-	4	1.93
*Exophiala*	2	3.85	1	3.70	0	-	0	-	0	-	3	1.45
*Microsporum*	3	5.77	0	-	0	-	0	-	0	-	3	1.45
*Monilia*	3	5.77	0	-	0	-	0	-	0	-	3	1.45
*Cryptococcus*	0	-	1	3.70	2	1.94	0	-	0	-	3	1.45
*Verticillium*	0	-	0	-	2	1.94	0	-	0	-	2	0.97
*Candida*	1	1.92	1	3.70	0	-	0	-	0	-	2	0.97
*Moniliella*	0	-	0	-	0	-	1	5.26	0	-	1	0.48
*Trichoderma*	1	1.92	0	-	0	-	0	-	0	-	1	0.48

**Table 2 pathogens-09-00783-t002:** Distribution of positive samples by type of ward. Intensive care unit was the type of ward with the highest frequency of positive samples, followed by the intermediate services, surgery wards, and medicine.

	Surgery (No = 91)		Intensive Care (No = 31)		Medicine (No = 209)		Intermediate Service (No = 23)		Others (No = 10)		Total (No = 364)	
	No	%	No	%	No	%	No	%	No	%	No	%
**Total samples**												
*Positive*	52	57.14	27	87.10	103	49.28	19	82.61	6	60.00	207	56.87
*Negative*	39	42.86	4	12.90	106	50.72	4	17.39	4	40.00	157	43.13
Molds												
*Positive*	49	53.85	23	74.19	98	46.89	19	82.61	4	40.00	195	53.57
*Negative*	42	46.15	8	25.81	111	53.11	4	17.39	6	60.00	169	46.43
Yeasts												
*Positive*	4	4.40	7	22.58	8	3.83	0	0.00	0	0.00	19	5.22
*Negative*	87	95.60	24	77.42	201	96.17	23	100.00	10	100.00	345	94.78

**Table 3 pathogens-09-00783-t003:** Distribution of positive samples by type room. Positive samples were obtained from hospital rooms, followed from bathrooms, and from operating theatres. The type of room was not statistically significant.

	Bathrooms (No = 66)		Hospital Rooms (No = 104)		Operatory Rooms (No = 73)		Others (No = 121)		Total (No = 364)	
	No	%	No	%	No	%	No	%	No	%
**Total samples**										
*Positive*	38	57.58	65	62.5	40	54.79	64	52.89	207	56.87
*Negative*	28	42.42	39	37.5	33	45.21	57	47.11	157	43.13
Molds research										
*Positive*	37	56.06	61	58.65	38	52.05	59	48.76	195	53.57
*Negative*	29	43.94	43	41.35	35	47.95	62	51.24	169	46.43
Yeasts research										
*Positive*	1	1.52	9	8.65	2	2.74	7	5.79	19	5.22
*Negative*	65	98.48	95	91.35	71	97.26	114	94.21	345	94.78

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
