# Peer review of "Occurrence of Fungi in the Potable Water of Hospitals: A Public Health Threat"

_pathogens, 2020, doi:10.3390/pathogens9100783_

Round 1

Reviewer 1 Report

The authors address an important quality parameter for potable water by collecting hospital samples and testing the presence of fungi. They find that filamentous fungi are more ubiquitous that yeast and the intensive care unit had the maximum burden. While the study is interesting, it will be great if the authors addressed the following comments to make their manuscript stronger:

  1. Figure 1 shows the fungi they observed in potable water. From the figure, it is a bit unclear how they did the CFU counts given the colonies are crowded. It will be great if the authors can add more information about this figure and about how they measured CFUs.
  2. It is unclear from the methods section how the different filamentous fungi were identified. The authors state that "filamentous fungi were identified phenotypically based on macroscopic and microscopic morphological characteristics" but it will be helpful if they provided more information. Filamentous fungi across different genus may look similar morphologically.
  3. Minor comment - for the period of sampling, it looks like the authors went by the calendar year. How do the data look if they were grouped by seasons, say, winter vs summer? 
  4. While the Odds Ratio seems like an interesting way to look at the data, the authors should provide more interpretation in the text for what those observations mean. Figure 2 plots overall need more explanation, for example, what's the round dot vs the diamond-shaped dot?
  5. Since some filamentous fungi such as Candida exist in yeast forms, it will be helpful if he authors describe what form they found and how they classified these dimorphic organisms. 

Author Response

We thank the Reviewer to for the  important comments to improve our manuscript, so we modified our paper according to his recommendations

  1. Figure 1 shows the fungi they observed in potable water. From the figure, it is a bit unclear how they did the CFU counts given the colonies are crowded. It will be great if the authors can add more information about this figure and about how they measured CFUs.

  Thanks for this comment, the figure 1 shows the fungi after eight days of incubation. We observed the plates every day, but the enumeration was evaluated at five days of incubation when the colonies growth was not crowded. We specified it in the text (line 277) and in the figure 1 caption.

 2. It is unclear from the methods section how the different filamentous fungi were identified. The authors state that "filamentous fungi were identified phenotypically based on macroscopic and microscopic morphological characteristics" but it will be helpful if they provided more information. Filamentous fungi across different genus may look similar morphologically.

We thank the Reviewer for this observation. We added this information in the text at the paragraph 4.3 (lines 280-289.)

  1. Minor comment for the period of sampling, it looks like the authors went by the calendar year. How do the data look if they were grouped by seasons, say, winter vs summer?

Thank for this comment, we considered the period of the year divided into three quarters (January to April, May to August, and September to December), to allow an analyze of time series

  1. While the Odds Ratio seems like an interesting way to look at the data, the authors should provide more interpretation in the text for what those observations mean. Figure 2 plots overall need more explanation, for example, what's the round dot vs the diamond-shaped dot?

We reported more explanation in the Figure 2 caption.

  1. Since some filamentous fungi such as Candida exist in yeast forms, it will be helpful if he authors describe what form they found and how they classified these dimorphic organisms.

Thanks for this important comment, Candida genus was observed only in yeast form and classified as yeast; however, in our study almost all samples were positive for moulds.

Reviewer 2 Report

General remarks:

The authors evaluated the occurrence of fungi in potable water collected from distribution systems in hospitals. 56.9% of samples were positive for fungi among them. The highest frequency of positive samples was found in intensive care unit. The period of the year was found to be statistically significant. The type of room did not have statistically significant effect, but type of departments had significant effect on fungal contamination. The manuscript is important in its subject matter, but it still needs some work before it can be published.

Corrections and remarks

To evaluate the environmental characteristics that could be involved in the contamination, the type of the departments (operative units) were classified by the authors as intensive care unit, medicine, surgery, intermediate services, and other services. The type of room, where the sample was collected, was classified as baths, hospital rooms, operative rooms, and others (management office, service rooms etc.). In the manuscript, these categories were not used consistently. The inconsistent use of terms makes the study difficult to understand. The term “operating unit” and “department” were used as synonyms. Do not use synonyms because it complicates the text. I suggest to use the term “department” in all cases. Similarly, the term “rooms” and “wards” are frequently replaced. Use “type of room” only, to provide a clear interpretation of the results.

Table 1: arrange results alphabetically, according fungal names

Row 82 and other places: write intensive care unit instead of ICU

row 208: “in few hospitals”: Please indicate, how many hospitals did you investigated?

Rows 192-193: explain better what is the relationship between corrosion and differences in residual chlorine in water distribution systems.

Minor corrections:

row 31 “distribute”, “d” is missing "

row 40 and Table 1 Mucorales- is not italic.

row 46: a dot is missing.

row 56 in Aspergillus A is capital letter

row 77 Candida is italic.

row 106 write one instead of 1

rows 113-115: please check grammar

Table 2: “Moulds research, Yeast research”, delete “research”.

Row 133 Use “opportunistic pathogen” instead of “occasionally pathogen”

Row 136 half of the samples

rows 161-164: please check grammar

rows 165-166: names of genera are italic

row 176: “environmental inhabitant” makes no sense.

row 177 delete “but”. Start new sentence.

Row 180: Please explain this better: “Although data on mould behaviour in the network are limited,”

Row 184: mention only relevant mycotoxin producers: Aspergillus, Fusarium and Penicillium.

Row 209 “flamed” instead of “flambéed”

Row 234 “sampling results” instead of “sample results”

Questions:

Did you analyse the difference of water quality among hospital buildings?

Do you have any information about the age or the last cleaning time of the pipelines of different hospital buildings?

Author Response

We thank the Reviewer to suggestions to improve our manuscript, so we modified our paper according to his recommendations

The inconsistent use of terms makes the study difficult to understand. The term “operating unit” and “department” were used as synonyms. Do not use synonyms because it complicates the text. I suggest to use the term “department” in all cases. Similarly, the term “rooms” and “wards” are frequently replaced. Use “type of room” only, to provide a clear interpretation of the results.

We agree with your important consideration so we always used the same term throughout the text

Table 1: arrange results alphabetically, according fungal names

Although we agree with the reviewer that the alphabetical order was more ordered, in our study the aim of Table 1 is to show the frequency of the fungal genera, so we reported the data in percentage in descending order

Row 82 and other places: write intensive care unit instead of ICU

Thank you, we modified “ICU” in intensive care unit

row 208: “in few hospitals”: Please indicate, how many hospitals did you investigated?

We investigated one hospital with different departments. We reported this information at line 226-227.

Rows 192-193: explain better what is the relationship between corrosion and differences in residual chlorine in water distribution systems.

 Thanks for your important consideration. We clarified this theme at line 206-210

Minor corrections:

Thank you, we reported the correction the text

Questions:

Did you analyse the difference of water quality among hospital buildings?

No, we didn’t analysed the difference of water quality among hospital buildings, our aim was to evaluate the occurrence of fungi in the potable water networks of hospitals. But we thank you for this important consideration and suggestion, we will plan to do this analysis in the future study.

Do you have any information about the age or the last cleaning time of the pipelines of different hospitals buildings?

Generally, in our hospital the disinfection processes of the water pipelines are performed on the basis of the results obtained from microbiological and legionella investigation. when the values ​​are beyond the threshold values, the disinfection takes place

Reviewer 3 Report

This paper is in general well written and easy to read. The methods must be improved. In particular, this question: is the Italian legislation different concerning the microbial quality of waters quite different from WHO recommendations??, and if so, what differences?. Should it be conflicts in legislation, these must be addressed in the paper.

Author Response

This paper is in general well written and easy to read. The methods must be improved. In particular, this question: is the Italian legislation different concerning the microbial quality of waters quite different from WHO recommendations??, and if so, what differences?. Should it be conflicts in legislation, these must be addressed in the paper.

We thanks the Reviewer for this question. The Italian legislation Decree 31/2001 follow the requirements of the European Communities, on the quality of water intended for human consumption